# How Biological Activity in Sea Cucumbers Changes as a Function of Species and Tissue

**DOI:** 10.3390/foods13010035

**Published:** 2023-12-21

**Authors:** Sabrina Sales, Helena M. Lourenço, Narcisa M. Bandarra, Cláudia Afonso, Joana Matos, Maria João Botelho, Maria Fernanda Pessoa, Pedro M. Félix, Arthur Veronez, Carlos Cardoso

**Affiliations:** 1Division of Aquaculture and Upgrading (DivAV), Portuguese Institute for the Sea and Atmosphere (IPMA, IP), Rua Alfredo Magalhães Ramalho 6, 1495-006 Lisbon, Portugal; sabrina.sales@ipma.pt (S.S.); helena@ipma.pt (H.M.L.); narcisa@ipma.pt (N.M.B.); cafonso@ipma.pt (C.A.); jojoanamatos7@gmail.com (J.M.); 2Division of Oceanography and Marine Environment (DivOA), Portuguese Institute for the Sea and Atmosphere (IPMA, IP), 1495-165 Lisbon, Portugal; mjbotelho@ipma.pt; 3GeoBioTec, Department of Earth Sciences, Faculty of Science and Technology (UNL), Largo da Torre, 2829-516 Caparica, Portugal; mfgp@fct.unl.pt; 4Interdisciplinary Centre of Marine and Environmental Research (CIIMAR), University of Porto, Rua dos Bragas 289, 4050-123 Porto, Portugal; 5MARE—Marine and Environmental Sciences Centre/ARNET—Aquatic Research Network, Faculty of Sciences, University of Lisbon, 1749-017 Lisbon, Portugal; pmfelix@fc.ul.pt; 6Centre for Functional Ecology—Science for People & the Planet, Department of Life Sciences, Faculty of Science and Technology, University of Coimbra, Calçada Martim de Freitas, 3000-456 Coimbra, Portugal; veronezarthuur@gmail.com

**Keywords:** holothurians, aqueous extracts, bioactive content, antioxidant, anti-inflammatory

## Abstract

Biological activity and bioactive compound content in sea cucumbers was assessed, considering *Parastichopus regalis*, *Holothuria mammata*, *Holothuria forskali*, and *Holothuria arguinensis* as species and intestine, muscle band, respiratory tree, body wall, and gonads as tissues. *P. regalis* had the lowest content in phenolic compounds and antioxidant activity in contrast to *Holothuria* species. In the respiratory tree, the highest phenolic concentration was recorded in *H. arguinensis*, 76.4 ± 1.2 mg GAE/100 g dw vs. 21.0–49.0 mg GAE/100 g dw in the other species. *H. arguinensis* had the highest DPPH and FRAP results in the gonads, 13.6 ± 0.7 mg AAE/100 g dw vs. 2.6–3.5 mg AAE/100 g dw and 27.1 ± 0.3 μmol Fe^2+^/g dw vs. 8.0–15.9 μmol Fe^2+^/g dw, respectively. Overall, *P. regalis* biomass presented the highest anti-inflammatory activity levels and *H. arguinensis* the lowest anti-inflammatory levels. The respiratory tree was the most anti-inflammatory (measured by the inhibition of cyclooxygenase-2, COX-2) tissue in *H. mammata* and *H. forskali* (also the muscle band in this case), 76.3 ± 6.3% and 59.5 ± 3.6% COX-2 inhibition in 1 mg/mL aqueous extracts, respectively. The results demonstrated a variable bioactive potential and advantage in targeting antioxidant properties in the muscle band and anti-inflammatory activity in the respiratory tree, which may constitute a starting point for a biorefinery approach envisaging multiple applications.

## 1. Introduction

Sea cucumbers are echinoderms, belonging to the class Holothuroidea, that perform important ecological tasks in the marine environment and are considered bio-indicators of water quality [1], including the presence of contaminants from natural sources or anthropogenic activities [2]. Their role as bio-indicators derives from them being deposit-feeders. A high nutritional potential has been ascribed to sea cucumbers [3]. Edible sea cucumbers are regarded as a culinary delicacy and an important dietary nutritional source by more than one billion consumers in the Asian continent [4,5]. In the last decades, sea cucumbers have gained a lot of attention from researchers due to their nutritional quality as well as their beneficial effect on human health [6,7]. The gamut of bioactive compounds identified in these animals is quite wide and comprises polysaccharides (including fucoidans), peptides, phospholipids, glycolipids (encompassing glycosphingolipids), phenols, and triterpene glycosides (saponins) [7,8,9,10].

Among the wide diversity of sea cucumbers [4,11], there are four relevant species in Portuguese waters that have been insufficiently studied: *Parastichopus regalis*, *Holothuria mammata*, *Holothuria forskali*, and *Holothuria arguinensis*. Some of these species, such as *H. forskali* [12], *H. mammata* [13], or *H. arguinensis* [14,15], have been acknowledged as commercially attractive and have already been studied for aquaculture purposes. Significant antioxidant activity was already identified in the biomass of *H. mammata*, *H. forskali*, and *H. arguinensis* [16,17]. Regarding *P. regalis*, a balanced nutritional quality has been identified [18], thus making its biomass suitable for human consumption. In particular, the biological activity found in the biomass of some sea cucumber species has been reported to reach high levels [16]. These authors found substantial anti-inflammatory activity in the ethanolic extracts attained from *H. mammata* and *H. arguinensis*, corresponding to an approximately 40% inhibition of the cyclooxygenase-2. In addition, significant levels of antioxidant activity in aqueous and ethanolic extracts have also been reported [16]. The presence of anti-inflammatory compounds, such as eicosapentaenoic acid (EPA), in sea cucumbers is also known [19]. Most studies do not consider specific tissues. A relevant exception is a recent study [20] which analysed the digestive tract, muscle, body wall, gonads, and the respiratory tree of *H. forskali* and reported significant antibacterial activity.

Considering all these aspects, an experimental design combining different holothurian tissues (intestine, muscle band, respiratory tree, body wall, and gonads) and four abundant species on the Portuguese coast (*P. regalis*, *H. mammata*, *H. forskali*, and *H. arguinensis*) was followed and all these samples were subjected to several analyses aimed at determining phenolic content and evaluating the antioxidant and anti-inflammatory activity of aqueous extracts.

## 2. Materials and Methods

### 2.1. Sample Source, Collection, and Preparation

Fresh samples of four species of sea cucumber, *Holothuria arguinensis*, *Holothuria forskali*, *Holothuria mammata*, and *Parastichopus regalis*, were captured in different coastal zones of Portugal between May 2022 and October 2022. *H. arguinensis* (*n* = 30), *H. forskali* (*n* = 30), and *H. mammata* (*n* = 30) were captured by scuba diving in three coastal areas of Setúbal: 38°24′43″ N 9°08′51″ W at an average depth of 15 m, 38°25′34″ N 9°06′22″ W at an average depth of 9 m, and 38°25′40″ N 9°03′25″ W at an average depth of 16 m. *P. regalis* (*n* = 30) were captured by a trawler in three areas: Aveiro (41°01′10″ N 9°51′40″ W) at an average depth of 150 m, Sintra-Cascais (38°50′26″ N 9°36′09″ W) at an average depth of 126 m, and Vicentine Coast (37°27′19″ N 8°57′10″ W) at a depth of 369 m. After being captured, still in water, the animals were placed in individual bags and brought quickly to IPMA’s laboratory in Lisbon in order to avoid the loss of their visceral content [21], since, when in stress, they can expel internal organs, a measure commonly used as a defence mechanism.

Sea cucumbers were then cleaned, washed, and correctly identified; phylogenetic identification was carried out in accordance with a previously described methodology [16]. The biometric data (total fresh weight; total length) were measured for a sample of 10 adult individuals of each species with similar biometric characteristics. To avoid excessive variability in measurement, resulting from muscle contraction and evisceration, a standardized procedure was followed, which was to weigh the entire contents of the bag and place the animals on ice to measure the length. Afterwards, the main tissues of the specimens were separated: intestine, muscle band, respiratory tree, body wall, and gonads. In the case of *H. forskali*, specimens were collected out of the breeding season and, for this reason, it was not possible to separate the gonad tissue.

Finally, the different sea cucumber tissues were then frozen, freeze-dried, ground to a powder, and stored at −80 °C until analysis.

### 2.2. Preparation of Extracts

Different extracts were prepared for the determination of the antioxidant (and polyphenol content) and anti-inflammatory activity.

For the phenolic compounds and antioxidant activity, an aqueous extraction was chosen, since this may facilitate the upgrading of specific tissue fractions for food, cosmetic or pharmaceutical purposes. In order to prepare the extracts, approximately 1.25 g of freeze-dried sea cucumber tissue was weighed, homogenized with 25 mL of water using a model Polytron PT 6100 homogenizer (Kinematica, Luzern, Switzerland) at a velocity of 30,000 rpm for 1 min, and agitated for 18 h using an orbital shaker. After centrifugation (5000× *g* at room temperature for 20 min), the supernatant was collected through a filter to a final volume of 25 mL.

Regarding the anti-inflammatory activity of the studied tissues of the sea cucumbers *P. regalis*, *H. mammata*, *H. forskali*, and *H. arguinensis*, aqueous extracts were attained in the following manner: approximately 200 mg of freeze-dried biomass or product was homogenized with 2 mL of water using a model Polytron PT6100 homogenizer (Kinematica, Luzern, Switzerland) at a velocity of 30,000 rpm for 1 min. The extracts were subjected to heat treatment (80 °C during 1 h) and then centrifuged (3000× *g* at 4 °C for 10 min). The supernatant was collected and the solvent was evaporated using a vacuum rotary evaporator with the water bath temperature at 65 °C. The residue was directly dissolved in 100% dimethyl sulfoxide (DMSO) to prepare a stock preparation with a concentration of 10 mg/mL.

### 2.3. Total Polyphenol Content

The total polyphenol content was determined in the prepared aqueous extracts according to the Singleton and Rossi method using the Folin–Ciocalteu reagent [22]. Briefly, 100 μL of extract was mixed with 600 μL ultra-pure water and 150 μL Folin–Ciocalteu reagent (diluted 1:1 with ultra-pure water) and, after 5 min dark incubation at room temperature, mixed with 750 μL 2% (*w*/*w*) sodium carbonate solution. The attained mixture was left for 1 h 30 min in the dark at room temperature; then, the absorbance was measured at 750 nm. Gallic acid (GA) was used as standard (based on a stock solution of 1 mg/mL gallic acid, solutions whose concentration was in the 0.01–0.30 mg/mL range were prepared and used for building a calibration curve) and phenolic content was expressed as gallic acid equivalents (mg GAE/100 g dw) through the calibration curve of gallic acid (Sigma, Steinheim, Germany).

### 2.4. Antioxidant Activity as Measured Using the DPPH Method

The antioxidant activity was measured through the determination of the radical scavenging activity using 2,2-diphenyl-1-picrylhydrazyl (DPPH) [23]. A volume of 1 mL of aqueous extract and 2 mL of DPPH (Sigma, Steinheim, Germany) 0.15 mM methanolic solution was thoroughly mixed. After 30 min of incubation at room temperature in the dark, absorbance was measured at 517 nm in a Helios Alpha model (Unicam, Leeds, UK) UV/visible light spectrophotometer. Water was used as the blank.

Radical scavenging activity was calculated using the following formula:% Inhibition = (A0 − Asample)/A0 × 100
where:A0—Absorbance of the blank; andAsample—Absorbance of the sample.

The results were expressed in mg of ascorbic acid equivalent (AAE) per 100 g dw and compared with a Trolox positive control (Trolox solution of 100 mg/L). The ascorbic acid calibration curve was attained by measuring the absorbance of methanolic solutions whose concentration was in the 5–25 mg/L range and prepared from a stock solution of 1 g/L ascorbic acid.

### 2.5. Antioxidant Activity as Measured Using the FRAP Method

The Ferric Ion Reducing Antioxidant Power (FRAP) method is a modified technique based on a previous method [24] and it was applied to the same extracts used in the previous sections. The results were expressed in μmol Fe^2+^ equivalent per g dw and compared with a Trolox-positive control (Trolox solution of 100 mg/L). The iron (II) sulphate calibration curve was obtained by determining the absorbance of solutions whose concentration was in the 0.07–0.60 mg/mL range and prepared from a stock solution of 1 mg/mL iron (II) sulphate.

### 2.6. Anti-Inflammatory Activity

The aqueous extract was tested at 1 mg/mL (after dilution of the stock solution) using a commercial cyclooxygenase (COX) inhibitory screening assay kit, Cayman test kit-560131 (Cayman Chemical Company, Ann Arbor, MI, USA). A volume of 10 μL each of test extract or DMSO (blank) was used. The results are expressed as a percentage of inhibition of COX-2 and compared with a diclofenac-positive control (diclofenac solution of 500 μg/L).

### 2.7. Statistical Analysis

Normality and variance homogeneity were tested using the Kolmogorov–Smirnov’s and Levene’s F-tests, respectively. The sea cucumber species and type of tissue were the studied factors through a two-way factorial ANOVA. The parametric test, Tukey HSD (unequal), was applied. All statistical treatment was carried out with STATISTICA 6, 2003 version (StatSoft, Inc., Tulsa, OK, USA), considering a significance level of (α) < 0.05.

## 3. Results and Discussion

### 3.1. Polyphenols and Antioxidant Activity

The total polyphenol content and the antioxidant activity (as measured by the DPPH and FRAP methods) in aqueous extracts of the studied sea cucumber species, *P. regalis*, *H. mammata*, *H. forskali*, and *H. arguinensis*, are presented in the Table 1, Table 2 and Table 3.

With respect to the total phenolic content, there were significant differences in the concentration of these bioactive compounds between species and sampled tissues. Without discriminating between tissues, *P. regalis* was the poorest in phenolic compounds in contrast to *H. mammata* and *H. arguinensis*. Indeed, if particular tissues are considered, this relative richness of *H. arguinensis* in phenolic substances is corroborated by the cases of the intestine, muscle band, and respiratory tree. For this latter tissue, the highest phenolic concentration was recorded in *H. arguinensis*, 76.4 ± 1.2 mg GAE/100 g dw vs. 21.0–49.0 mg GAE/100 g dw in the other three species. For *H. mammata* gonads, a high total phenolic content surpassing the other studied sea cucumber species was observed, 60.4 ± 0.5 mg GAE/100 g dw vs. 15.4–49.9 mg GAE/100 g dw. On the other hand, an analysis of the results only discriminating between specific tissues is also possible. This analysis showed that the muscle band was the richest tissue in phenolic substances, thus contrasting with the intestine, the poorest tissue regarding this parameter. In fact, if particular species are considered, the muscle band exhibited the highest values in both *P. regalis* and *H. forskali*, while the intestine had the lowest values in *H. mammata* and *H. forskali* —the Cuvierian tubules of this species were also analysed and were equally poor in phenolics, 6.2 ± 1.1 mg GAE/100 g dw. Moreover, other relatively rich tissues, such as the gonads and respiratory tree, presented the highest contents of all tissues in one species each: 60.4 ± 0.5 mg GAE/100 g dw vs. 15.9–56.0 mg GAE/100 g dw in *H. mammata* and 76.4 ± 1.2 mg GAE/100 g dw vs. 28.0–62.5 mg GAE/100 g dw in *H. arguinensis*, respectively.

The DPPH determination of the antioxidant activity revealed a different quantitative relationship between species and tissues. However, some trends were similar to those already observed for the phenolic content, namely, if averages across all species are calculated and statistically analysed, the muscle band was the most antioxidant tissue, thus matching its phenolic concentration. *P. regalis* biomass (averaging all tissues) was the least antioxidant as measured by DPPH, thereby also matching its very poor phenolic levels. At a more detailed level, an analysis of the results shows that the intestine, muscle band, and respiratory tree of *P. regalis* was the least antioxidant set of tissues if compared with the same tissues of the other three species. For instance, *P. regalis* intestine presented a DPPH value of only 1.9 ± 0.5 mg AAE/100 g dw, which compares to 8.4–15.1 mg AAE/100 g dw in the other sea cucumbers. *H. arguinensis* had the most antioxidant intestinal tissue —belonging to the highest DPPH values—and also the highest DPPH result among the gonads, 13.6 ± 0.7 mg AAE/100 g dw vs. 2.6–3.5 mg AAE/100 g dw. In the case of the body wall, no difference was observed between species. Furthermore, the muscle band was the most antioxidant tissue in two species, *H. forskali* and *H. mammata* (together with the respiratory tree), reaching 13.9 ± 0.6 mg AAE/100 g dw and 11.3 ± 0.9 mg AAE/100 g dw, respectively. The positive control (Trolox solution at 100 mg/L) yielded DPPH results similar to those of the intestine and gonads in *H. arguinensis* (less than 15% deviation).

The antioxidant activity measured using the other method (FRAP) did not yield results that deviated much from those determined with the DPPH method. Indeed, *P. regalis* biomass was overall the least antioxidant and the muscle band was the most antioxidant tissue. *P. regalis* was always among the least antioxidant species in the cases of the intestine, muscle band, respiratory tree, and body wall. In two of them, intestine and respiratory tree, *P. regalis* was the least antioxidant species of all as measured by the FRAP. For the gonads, *H. mammata* displayed the least antioxidant activity, 8.0 ± 0.3 μmol Fe^2+^/g dw vs. 15.9–27.1 μmol Fe^2+^/g dw. If the highest values are considered, *H. forskali* and *H. arguinensis* must be highlighted. The former had the most antioxidant muscle band and body wall, 28.4 ± 0.4 μmol Fe^2+^/g dw and 20.0 ± 0.1 μmol Fe^2+^/g dw, respectively, and the latter the most antioxidant intestine and gonads, 28.5 ± 0.6 μmol Fe^2+^/g dw and 27.1 ± 0.3 μmol Fe^2+^/g dw, respectively. With the exception of *H. arguinensis*, the muscle band was always the most antioxidant tissue as measured by FRAP, 23.5–28.4 μmol Fe^2+^/g dw. The respiratory tree was the least antioxidant tissue in *P. regalis* and *H. forskali*. The positive control (Trolox solution at 100 mg/L) yielded FRAP results similar to those of the muscle band in the studied species (less than 15% deviation).

Although these marine organisms have not been deeply and comprehensively studied, there are some studies on antioxidant activity in sea cucumbers, but not necessarily on the four species selected for the current study [11,16,17,25,26]. Concerning phenolic content, there is a recent study [16] that sampled three of the four sea cucumber species, *H. mammata*, *H. forskali*, and *H. arguinensis*, and also prepared aqueous extracts. Phenolic levels in the 48–84 mg GAE/100 g dw range in the whole biomass of these species have been determined [16], thereby overlapping with the values observed in the muscle band and respiratory tree. The values for *H. forskali* were at the lower end of the interval [16]. Much higher values in biomass derived from *H. forskali*, >100 mg GAE/100 g dw, have been reported by other authors [17]. Other species of the *Holothuria* genus have also been studied, such as *H. scabra*, which had very high phenolic levels exceeding 2000 mg GAE/100 g dw, but in the case of organic solvent extracts [25,27].

The differentiation between tissues is absent from the aforementioned studies. A recent study [28] on internal organs (gonads, respiratory tracts, and intestines) of *Cucumaria frondosa* found 150–250 mg GAE/100 g dw. However, besides *C. frondosa* being a suspension-feeder, unlike the species in this study, reference [28] does not adequately separate between different tissues. In any case, the Canadian sea cucumber study [28] identified catechin, cinnamic acid, gallic acid, hydroxygallic acid, protocatechuic acid, and quercetin as the main phenolic compounds in the internal organs. The most detailed study, to the authors’ knowledge, was also on *C. frondosa* [29] and found high total phenolic contents in the acetonitrile-rich fractions of the digestive tract (236 mg GAE/100 g dw), respiratory apparatus (200 mg GAE/100 g dw), muscle (194 mg GAE/100 g dw), and gonads (130 mg GAE/100 g dw), thereby surpassing the values in the water-rich fractions. The latter were within the current study’s range. In purely aqueous extractions, the highest values (112 mg GAE/100 g dw) have been found in the muscle tissues [29]. The presence of easily assimilated antioxidant phenols, such as anthocyanins, anthocyanidins, tannins, and others in the food sources of the sea cucumber were proposed as their source in the studied tissues. However, further research on this subject (especially given the outstanding levels found in muscle tissues) is warranted.

Concerning antioxidant activity as measured by DPPH, researchers working with the whole biomass did not detect any activity in the aqueous extracts from *H. arguinensis*, but observed high activity levels in the aqueous extracts from *H. mammata* and *H. forskali*, >30 mg AAE/100 g dw [16]. This contrasts with the current study’s results, since *H. arguinensis* tissues had some of the highest DPPH results. The harvesting season may be an explanation for these differences. High DPPH inhibition levels have been reported for other species, such as 82–95% in a phosphate buffer extract from the *H. atra* body wall [30]. This is equivalent to less than 40% inhibition in the case of the sea cucumbers of the present study, in which the body wall levels were particularly low. However, a study on the antioxidant potential in the coelomic fluid of three sea cucumber species (of two genera, *Bohadschia* and *Stichopus*) collected from Malaysian coastal waters reported DPPH inhibition values lower than 52%, thus also corresponding to a moderate radical scavenging activity [31].

Finally, as for FRAP, a first comparison to the literature suggests relatively high antioxidant activity in the four studied species, since FRAP values of 6.2–8.9 μmol Fe^2+^/g dw in the aqueous extracts from *H. mammata*, *H. forskali*, and *H. arguinensis* were measured [16]. These are whole biomass values that are only matched by low antioxidant FRAP levels in the intestine and respiratory tree of *P. regalis*, as well as the gonads of *H. mammata* (Table 3). More relevant FRAP activity levels were reported for dermal extracts from *H. tubulosa* [32].

It is known that the phenolic content may correlate with the antioxidant activity in the extracts, since the structural characteristics of phenolic molecules make strong antioxidant action possible whenever high contents of phenolics occur [33]. Hence, the correlations between total phenolic content and either DPPH or FRAP were analysed. However, correlations had an R^2^ of 0.13 for DPPH (R^2^ of 0.46 if only muscle tissues are considered) and 0.11 for FRAP. On the other hand, given the striking similarity between FRAP and DPPH results, the correlation between these two sets of antioxidant results could be assessed and shown to correspond to a R^2^ of 0.85 (Figure 1). Hence, there is a high level of agreement between two distinct methods, which quantify different aspects of the antioxidant activity. In fact, while the FRAP test is related to the presence of potential electron donors, the DPPH assay is used for the determination of molecules with proton (H•^−^) donor potential, being associated with a more complex underlying mechanism [34]. In accordance, antioxidant compounds other than phenolic substances may play a major role in determining antioxidant activity in the various tissues of the sea cucumbers, though phenolic substances may be strong contributors to the antioxidant activity in holothurian muscle tissues. It is also possible that specific phenolic compounds play major roles, and that higher antioxidant activities are thus determined by particular phenolic profiles [30,35,36] and not by a high overall total of phenolic substances. For instance, antioxidant activity, such as that measured using FRAP, was related to the total flavonoid (a specific class of phenolic compounds) content in *H. scabra* [37]. Moreover, other classes of compounds may have a role in the antioxidant activity of sea cucumbers. For instance, *C. frondosa*, a source of very potent triterpene glycosides, contains a complex of EPA-enriched phospholipids that was associated with a marked reduction in hydrogen-peroxide-induced oxidative damage in the rat adrenal pheochromocytoma cell line PC12 [7,10].

### 3.2. Anti-Inflammatory Activity

The anti-inflammatory activity (expressed as a percentage of inhibition of COX-2) in the aqueous extracts of the selected sea cucumber species *P. regalis*, *H. mammata*, *H. forskali*, and *H. arguinensis* is shown in Table 4.

An overall analysis based on the averages of all analysed tissues indicated that *P. regalis* biomass was the most anti-inflammatory and *H. arguinensis* biomass the least anti-inflammatory. Additionally, if all species are averaged, the respiratory tree showed the highest anti-inflammatory activity, thereby contrasting with the body wall. A more fine-grained analysis enables the identification of *P. regalis* as the most anti-inflammatory species in the cases of the gonads and intestines (alongside *H. mammata*). In particular, a very high activity was registered in *P. regalis* gonads, 94.6 ± 4.0% COX-2 inhibition. *H. mammata* also had a high anti-inflammatory activity in its respiratory tree, the highest value in comparison to the same tissue of other species, 76.3 ± 6.3% COX-2 inhibition vs. 20.1–59.5% COX-2 inhibition. Regarding the muscle band, *H. forskali* had the highest COX-2 inhibition, 67.2 ± 2.5%. As for the body wall, values were all low and did not differ between sea cucumber species. On the other hand, the respiratory tree was the tissue that had high activity levels more consistently across species, being the most anti-inflammatory tissue in *H. mammata* and *H. forskali* (together with the muscle band in this case), 76.3 ± 6.3% COX-2 inhibition and 59.5 ± 3.6% COX-2 inhibition, respectively. The buccal apparatus of *P. regalis* was also analysed and showed moderate anti-inflammatory activity, 22.8 ± 5.3% COX-2 inhibition. The diclofenac solution (500 μg/L) exhibited 68.9 ± 1.5% COX-2 inhibition.

The aqueous extracts of the various tissues and species displayed a wide variability in the anti-inflammatory activity levels, from undetected to almost 95% COX-2 inhibition. This is a quite meaningful inhibition, since the tested positive control (a 500 μg/L diclofenac solution) did not reach such a high value. In any case, it is worth noting that the observed variability differs from the results for the aqueous extracts from *H. mammata*, *H. forskali*, and *H. arguinensis* using the same in vitro methodology for assessing the anti-inflammatory activity as in the current study, since they showed no COX-2 inhibition [16]. Besides seasonal effects, it is possible that tissues with low activity, such as the body wall, outweighed tissues with high activity, such as the respiratory tree, in the whole biomass used [16]. Other research groups have also investigated anti-inflammatory activity in holothurian extracts, but through different methods, such as inducing inflammation in THP-1 cells [38], nitric oxide production inhibition [27] or using in vivo models [39].

Other relevant anti-inflammatory activities of holothurian extracts have been previously observed. Results supporting the existence of promising anti-inflammatory activities have been observed in *H. polii* water–ethanol extracts [38]. Specifically, a derived aqueous fraction from the whole biomass was able to reduce the levels of the inflammatory markers interleukin-6, nitric oxide, matrix metalloproteinase 9 in the mouse mammary SCp2 cells, and the level of interleukin-1β generated by phorbol-12-myristate-13-acetate-activated THP-1 human monocytic cells [38]. For another sea cucumber of the *Holothuria* genus (*H. scabra*), an ethyl acetate fraction from the whole biomass was shown to modulate inflammation in vitro through the inhibition of nitric oxide and pro-inflammatory cytokines production through NF-κB and JNK pathways [27]. Another study on *H. scabra* [40] revealed that a bathing method applied to a fish tissue using an organic solvent holothurian extract concentration at 100 mg/L was optimal for spleen and gill protection against *Aeromonas hydrophila* bacteria, thereby inferring an anti-inflammatory action. All these studies only considered the whole biomass of the respective sea cucumber, thus strengthening the novelty of the current study.

The high anti-inflammatory activity of a specific tissue such as the respiratory tree or digestive tract may be ascribed either to the specific compounds biosynthesised and accumulated in that tissue or to aspects of the microbiome [41] affecting local biochemistry. However, the bioactive compounds underlying the observed anti-inflammatory activity were not identified. The phenolic substances reported in the four selected holothurian species are also a possibility. Therefore, a possible correlation between total phenolic content and anti-inflammatory activity was assessed, but R^2^ did not exceed 0.10. Moreover, to the best of the authors’ knowledge, no previous investigation of the anti-inflammatory activity of phenolic substance in sea cucumbers has been done. Other water-soluble compounds with anti-inflammatory properties may be present in these marine organisms [38]. Saponins are also a possibility [42], since triterpene glycosides (alongside terpenes and steroids) are major compounds in different groups of echinoderms [43,44]. These molecules act as a powerful aposematic signal for sea cucumbers against predators (warning potential predators about the unpalatability of the holothurian tissues) and are associated with internal organs (namely, respiratory tree, intestine, Cuvierian tubules, and gonads) that can be expelled as a defense mechanism [45,46].

Moreover, it should be noted that the identification of the respiratory tree or digestive tract as tissues with high concentrations of anti-inflammatory substances may constitute a starting point for further studies supporting a biorefinery approach, since other specific extracts and further fractioning of the aqueous extracts attained from these tissues may enable the extraction of highly biologically active products with nutraceutical or pharmacological applications, while generating other fractions and leaving other tissues with different properties, such as antioxidant activity, suitable to be used as macro- and micronutrient sources by the food industry.

## 4. Conclusions

Regarding antioxidant properties, the aqueous extracts of the four studied sea cucumbers and various tissues displayed large significant differences in the phenolic contents and DPPH and FRAP levels. A relative richness of *H. arguinensis* in phenolic substances was observed in the cases of the intestine, muscle band, and respiratory tree. Moreover, *H. arguinensis* had the most antioxidant intestinal tissue and also the highest DPPH and FRAP results in the gonads, 13.6 ± 0.7 mg AAE/100 g dw vs. 2.6–3.5 mg AAE/100 g dw and 27.1 ± 0.3 μmol Fe^2+^/g dw vs. 8.0–15.9 μmol Fe^2+^/g dw, respectively. With respect to the anti-inflammatory properties, *P. regalis* biomass was the most anti-inflammatory, and *H. arguinensis* biomass was the least anti-inflammatory. In general, the respiratory tree showed the highest anti-inflammatory activity, thereby contrasting with the body wall. In particular, the respiratory tree was the most anti-inflammatory tissue in *H. mammata* and *H. forskali* (together with the muscle band in this case), 76.3 ± 6.3% COX-2 inhibition and 59.5 ± 3.6% COX-2 inhibition, respectively. Therefore, the results showed a variable bioactive potential in the sea cucumber species and an advantage in targeting antioxidant properties in the muscle band and anti-inflammatory activity in the respiratory tree, which may constitute a starting point for further studies supporting a biorefinery approach envisaging nutraceutical, cosmetic or even pharmaceutical applications. In addition, future studies should evaluate the bioaccessibility of the biologically active compounds in sea cucumber (and in derived applications, such as functional foods) as well as assess the effects of drying, processing, and storage on the properties of tissues and related products.

## Figures and Tables

**Figure 1 foods-13-00035-f001:**
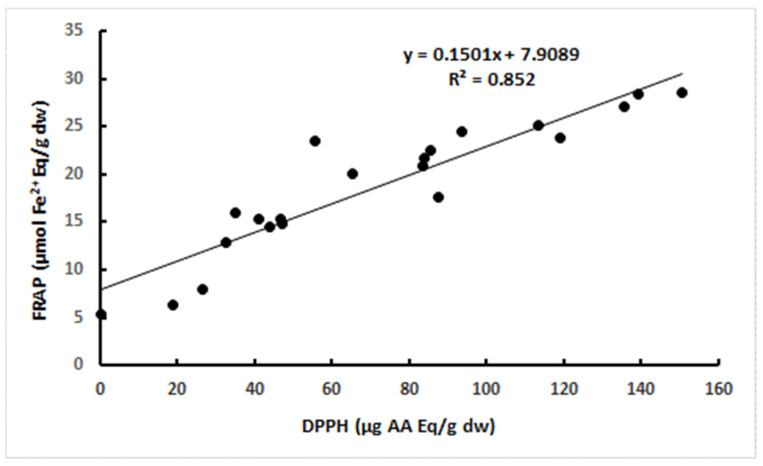
Correlation between the antioxidant results derived from the application of the DPPH and FRAP methods to all sea cucumber samples.

**Table 1 foods-13-00035-t001:** Total polyphenol content (in mg GAE/100 g dry weight) in aqueous extracts of selected tissues (intestine, muscle band, respiratory tree, body wall, and gonads) belonging to the studied sea cucumber species *Parastichopus regalis*, *Holothuria mammata*, *Holothuria forskali*, and *Holothuria arguinensis*.

Tissue	Total Polyphenol (mg GAE/100 g dw)			
	*Parastichopus regalis*	*Holothuria mammata*	*Holothuria forskali*	*Holothuria arguinensis*
Intestine	18.8 ± 0.0 ^bA^	15.9 ± 0.5 ^cA^	9.7 ± 0.8 ^dA^	37.5 ± 0.3 ^aB^
Muscle band	29.8 ± 0.6 ^aB^	56.0 ± 1.4 ^cD^	53.1 ± 0.9 ^bD^	62.5 ± 0.8 ^dD^
Respiratory tree	21.0 ± 0.8 ^aA^	49.0 ± 0.5 ^bC^	48.5 ± 0.3 ^bC^	76.4 ± 1.2 ^cE^
Body Wall	21.6 ± 0.6 ^aA^	32.4 ± 0.6 ^cB^	37.3 ± 0.5 ^dB^	28.0 ± 0.3 ^bA^
Gonads	15.4 ± 0.8 ^aA^	60.4 ± 0.5 ^cE^	NA	49.9 ± 0.5 ^bC^

Values are presented as average ± standard deviation. NA—Not analysed. Different lowercase letters within a row correspond to statistical differences (*p* < 0.05) between species. Different uppercase letters within a column correspond to statistical differences (*p* < 0.05) between tissues.

**Table 2 foods-13-00035-t002:** Antioxidant activity measured by the DPPH (in mg AAE/100 g dw) in aqueous extracts of selected tissues (intestine, muscle band, respiratory tree, body wall, and gonads) belonging to the studied sea cucumber species *Parastichopus regalis*, *Holothuria mammata*, *Holothuria forskali*, and *Holothuria arguinensis*.

Tissue	DPPH (mg AAE/100 g dw)			
	*Parastichopus regalis*	*Holothuria mammata*	*Holothuria forskali*	*Holothuria arguinensis*
Intestine	1.9 ± 0.5 ^aA^	8.5 ± 0.7 ^bB^	8.4 ± 1.5 ^bB^	15.1 ± 0.8 ^cC^
Muscle band	5.6 ± 0.1 ^aB^	11.3 ± 0.9 ^bC^	13.9 ± 0.6 ^cC^	9.4 ± 0.1 ^bB^
Respiratory tree	ND ^aA^	11.9 ± 0.8 ^dC^	3.3 ± 1.2 ^bA^	8.4 ± 0.6 ^cB^
Body Wall	4.7 ± 0.3 ^aB^	4.7 ± 1.0 ^aA^	6.5 ± 1.0 ^aB^	4.1 ± 1.0 ^aA^
Gonads	3.5 ± 0.2 ^aB^	2.6 ± 0.9 ^aA^	NA	13.6 ± 0.7 ^bC^

Values are presented as average ± standard deviation. ND—Not detected. NA—Not analysed. Different lowercase letters within a row correspond to statistical differences (*p* < 0.05) between species. Different uppercase letters within a column correspond to statistical differences (*p* < 0.05) between tissues.

**Table 3 foods-13-00035-t003:** Antioxidant activity measured by the FRAP (in μmol Fe^2+^/g dw) method in aqueous extracts of selected tissues (intestine, muscle band, respiratory tree, body wall, and gonads) belonging to the studied sea cucumber species *Parastichopus regalis*, *Holothuria mammata*, *Holothuria forskali*, and *Holothuria arguinensis*.

Tissue	FRAP (μmol Fe^2+^/g dw)			
	*Parastichopus regalis*	*Holothuria mammata*	*Holothuria forskali*	*Holothuria arguinensis*
Intestine	6.4 ± 0.1 ^aB^	22.5 ± 0.3 ^bC^	21.7 ± 0.2 ^bC^	28.5 ± 0.6 ^cE^
Muscle band	23.5 ± 0.3 ^aD^	25.1 ± 0.4 ^bE^	28.4 ± 0.4 ^cD^	24.5 ± 0.0 ^abC^
Respiratory tree	5.4 ± 0.2 ^aA^	23.8 ± 0.4 ^dD^	12.9 ± 0.4 ^bA^	20.9 ± 0.3 ^cB^
Body Wall	15.2 ± 0.1 ^aC^	14.8 ± 0.4 ^aB^	20.0 ± 0.1 ^bB^	15.3 ± 0.4 ^aA^
Gonads	15.9 ± 0.1 ^bC^	8.0 ± 0.3 ^aA^	NA	27.1 ± 0.3 ^cD^

Values are presented as average ± standard deviation. NA—Not analysed. Different lowercase letters within a row correspond to statistical differences (*p* < 0.05) between species. Different uppercase letters within a column correspond to statistical differences (*p* < 0.05) between tissues.

**Table 4 foods-13-00035-t004:** Anti-inflammatory activity (% inhibition of COX-2) in aqueous extracts of selected tissues (intestine, muscle band, respiratory tree, body wall, and gonads) belonging to the studied sea cucumber species, *Parastichopus regalis*, *Holothuria mammata*, *Holothuria forskali*, and *Holothuria arguinensis*.

Tissue	Anti-Inflammatory Activity (% COX-2 Inhibition)			
	*Parastichopus regalis*	*Holothuria mammata*	*Holothuria forskali*	*Holothuria arguinensis*
Intestine	29.8 ± 1.5 ^aB^	31.6 ± 5.8 ^aB^	13.1 ± 4.4 ^bA^	15.5 ± 6.0 ^bA^
Muscle band	2.6 ± 0.0 ^bA^	13.1 ± 16.3 ^bA^	67.2 ± 2.5 ^cB^	ND ^aB^
Respiratory tree	58.4 ± 4.6 ^bC^	76.3 ± 6.3 ^aC^	59.5 ± 3.6 ^bB^	20.1 ± 12.9 ^cA^
Body Wall	12.7 ± 1.6 ^aA^	2.3 ± 0.0 ^aA^	7.8 ± 0.0 ^aA^	6.8 ± 2.3 ^aA^
Gonads	94.6 ± 4.0 ^aD^	10.2 ± 6.4 ^bA^	NA	ND ^cB^

Values are presented as average ± standard deviation. ND—Not detected. NA—Not analysed. Different lowercase letters within a row correspond to statistical differences (*p* < 0.05) between species. Different uppercase letters within a column correspond to statistical differences (*p* < 0.05) between tissues.

## Data Availability

Data is contained within the article.

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
