# Peer review of "How Biological Activity in Sea Cucumbers Changes as a Function of Species and Tissue"

_foods, 2023, doi:10.3390/foods13010035_

Round 1
Reviewer 1 Report
Comments and Suggestions for Authors
Some revisions:
1. Please do not combine the different citation types (lines 55, 60, 64, 94, 146, 253, 262, 272, 278, 286, 292, 297, 316, 360, 365, 370)
2. How many samples of P. regalis were used? The other species was n =30.
3. Please explain why the gonad of H. forskali was not analyzed (NA in the table).
4. What does "similar" mean in lines 231 & 247?
5. What is the importance of comparing C. frondosa to the other 4 samples (line 261 - 273)?
6. Please delete the title at the top of figure 1 because there is already a title at the bottom. (line 323)
7. Please explain why the correlation between DPPH and FRAP is "very poor" (line 302) and "equally poor" (line 391)

Author Response
Dear Sir,
The whole paper was subjected to a new assessment in order to address the reviewer (and editor) concerns (including those in the annotated PDF file). For your advantage, we send you the revised manuscript with all alterations properly highlighted.
Reviewer #1:
Some revisions:
- Please do not combine the different citation types (lines 55, 60, 64, 94, 146, 253, 262, 272, 278, 286, 292, 297, 316, 360, 365, 370)
You are right and this issue was tackled in the manuscript’s text.
- How many samples of P. regalis were used? The other species was n =30.
Thank you for detecting this problem. The missing information was included in the text.
- Please explain why the gonad of H. forskali was not analyzed (NA in the table).
This explanation was added to the paper’s text. The reason was that this species’ samples were not collected in the breeding season.
- What does "similar" mean in lines 231 & 247?
The quantitative meaning of this word was included in the manuscript’s text.
- What is the importance of comparing C. frondosa to the other 4 samples (line 261 - 273)?
It is another relatively well studied species of sea cucumber.
- Please delete the title at the top of figure 1 because there is already a title at the bottom. (line 323)
Thank you. Figure 1 was changed accordingly.
- Please explain why the correlation between DPPH and FRAP is "very poor" (line 302) and "equally poor" (line 391)
This related to the determined values of R2. The terminology was revised and we abstained from qualifying adjectives.
We sincerely hope that the performed alterations are clear and acceptable. However, if needed, we will gladly take the changes in the manuscript even further.
With our best regards,
The Authors
Reviewer 2 Report
Comments and Suggestions for Authors
The research article is intriguing, but I would advise against publishing it in its current form due to several areas that require substantial improvement. Here are some specific comments:
General Comments:
- The English writing needs further improvement, as there are numerous grammatical or typographical errors throughout the manuscript. It is recommended to seek the assistance of a native speaker to refine the language.
Material and Methods:
- The Materials and Methods section should provide more detailed information, particularly regarding the specific standards used (e.g., the percentage applied).
Units:
- Throughout the entire manuscript, "ml" should be written as "mL," and "µl" should be written as "µL" to adhere to the standard unit conventions.
Minor Comments:
Line 47 adds a comma before and
Line 52 adds a comma before or
Line 64 changes the study to a study
Line 65 adds the before respiratory tree
Line 74 changes methods to Methods
Line 93 adds and before correctly
Line 94 changes to the methodology to with the methodology
Line 95 changes afterwards to afterward
Lines 106 and 107 add for before during
Line 114 adds for before during
Line 109 is it key tissues or issue ??
Line 130 changes were to was
Line 140 changes torolox to Torolox and mg/l to mg/L
line 147 changes torolox to Torolox and mg/l to mg/L
line 151 changes mg/ml to mg/mL
line 153 changes μl to μL
line 163 is it (α) 0.05. or < 0.05.
line 212 removes the comma after tree
line 220 change A more to More
line 222 change were to was
line 249 change deep to deeply
line 256 change in the lower to at the lower
line 272 removes in
line 277 removes the comma after tissues
line 281 changes This contrast with current to This contrasts with the current
line 285 change in the sea to of the sea
line 305 changes was to were
line 307 adds the before FRAP
line 309 changes to a more to with amore
line 320 changes to a marked to with a marked
line 321 removes in before hydrogen
line 327 adds a before percentage
line 339 removes the comma after tissues
line 354 removes a before moderate
line 359 removes the comma after solution
line 385 removes the comma after tract
line 398 changes to to with
line 399 changes are to is
line 410 adds a comma before and
line 411 adds was before the least
line 416 adds an or the before advantage
Comments on the Quality of English Language
The research article is intriguing, but I would advise against publishing it in its current form due to several areas that require substantial improvement. Here are some specific comments:
General Comments:
- The English writing needs further improvement, as there are numerous grammatical or typographical errors throughout the manuscript. It is recommended to seek the assistance of a native speaker to refine the language.
Material and Methods:
- The Materials and Methods section should provide more detailed information, particularly regarding the specific standards used (e.g., the percentage applied).
Units:
- Throughout the entire manuscript, "ml" should be written as "mL," and "µl" should be written as "µL" to adhere to the standard unit conventions.
Minor Comments:
Line 47 adds a comma before and
Line 52 adds a comma before or
Line 64 changes the study to a study
Line 65 adds the before respiratory tree
Line 74 changes methods to Methods
Line 93 adds and before correctly
Line 94 changes to the methodology to with the methodology
Line 95 changes afterwards to afterward
Lines 106 and 107 add for before during
Line 114 adds for before during
Line 109 is it key tissues or issue ??
Line 130 changes were to was
Line 140 changes torolox to Torolox and mg/l to mg/L
line 147 changes torolox to Torolox and mg/l to mg/L
line 151 changes mg/ml to mg/mL
line 153 changes μl to μL
line 163 is it (α) 0.05. or < 0.05.
line 212 removes the comma after tree
line 220 change A more to More
line 222 change were to was
line 249 change deep to deeply
line 256 change in the lower to at the lower
line 272 removes in
line 277 removes the comma after tissues
line 281 changes This contrast with current to This contrasts with the current
line 285 change in the sea to of the sea
line 305 changes was to were
line 307 adds the before FRAP
line 309 changes to a more to with amore
line 320 changes to a marked to with a marked
line 321 removes in before hydrogen
line 327 adds a before percentage
line 339 removes the comma after tissues
line 354 removes a before moderate
line 359 removes the comma after solution
line 385 removes the comma after tract
line 398 changes to to with
line 399 changes are to is
line 410 adds a comma before and
line 411 adds was before the least
line 416 adds an or the before advantage
Author Response
Dear Sir,
The whole paper was subjected to a new assessment in order to address the reviewer (and editor) concerns. For your advantage, we send you the revised manuscript with all alterations properly highlighted.
Reviewer #2:
The research article is intriguing, but I would advise against publishing it in its current form due to several areas that require substantial improvement. Here are some specific comments:
Thank you for your appraisal. We tried to improve it and respond to your specific criticisms below.
General Comments:
- The English writing needs further improvement, as there are numerous grammatical or typographical errors throughout the manuscript. It is recommended to seek the assistance of a native speaker to refine the language.
The whole manuscript was revised by someone whose mother tongue is English.
Material and Methods:
- The Materials and Methods section should provide more detailed information, particularly regarding the specific standards used (e.g., the percentage applied).
Thank you. Additional information concerning the specific standards was included in the manuscript’s text.
Units:
- Throughout the entire manuscript, "ml" should be written as "mL," and "µl" should be written as "µL" to adhere to the standard unit conventions.
You are right, it was done as such throughout the whole document.
Minor Comments:
Line 47 adds a comma before and
Done as requested by the reviewer.
Line 52 adds a comma before or
Done as asked.
Line 64 changes the study to a study
Done as asked.
Line 65 adds the before respiratory tree
Done as demanded by the reviewer.
Line 74 changes methods to Methods
Done as requested.
Line 93 adds and before correctly
Done as asked by the reviewer.
Line 94 changes to the methodology to with the methodology
Done as requested.
Line 95 changes afterwards to afterward
Done as asked.
Lines 106 and 107 add for before during
Done as proposed by the reviewer.
Line 114 adds for before during
Done as requested.
Line 109 is it key tissues or issue ??
The text was amended (it is tissue).
Line 130 changes were to was
Done as asked.
Line 140 changes trolox to Trolox and mg/l to mg/L
Both changes were done. Thank you for detecting these errors. Amendments were done in similar instances throughout the document.
line 147 changes trolox to Trolox and mg/l to mg/L
Done as requested.
line 151 changes mg/ml to mg/mL
Done as demanded by the reviewer.
line 153 changes μl to μL
Done as asked.
line 163 is it (α) 0.05. or < 0.05.
You are right, this was corrected in the manuscript’s text.
line 212 removes the comma after tree
Removed as requested by the reviewer.
line 220 change A more to More
Text was changed.
line 222 change were to was
Done as requested.
line 249 change deep to deeply
Changed as requested.
line 256 change in the lower to at the lower
Done as asked by the reviewer.
line 272 removes in
Done as demanded.
line 277 removes the comma after tissues
Done as asked.
line 281 changes This contrast with current to This contrasts with the current
You are right. Text was corrected.
line 285 change in the sea to of the sea
Done as requested and text was modified.
line 305 changes was to were
Text was changed.
line 307 adds the before FRAP
Done as proposed by the reviewer.
line 309 changes to a more to with amore
Done as requested.
line 320 changes to a marked to with a marked
Altered as defended by the reviewer.
line 321 removes in before hydrogen
You are right and this error was corrected.
line 327 adds a before percentage
Done as asked.
line 339 removes the comma after tissues
Done as demanded by the reviewer.
line 354 removes a before moderate
Done as requested.
line 359 removes the comma after solution
Done as asked.
line 385 removes the comma after tract
This comma was also removed.
line 398 changes to to with
Done as requested.
line 399 changes are to is
Text was modified.
line 410 adds a comma before and
Done as recommended by the reviewer.
line 411 adds was before the least
You are right, it is missing. Text was corrected.
line 416 adds an or the before advantage
The word “an” was added to the text.
We sincerely hope that the performed alterations are clear and acceptable. However, if needed, we will gladly take the changes in the manuscript even further.
With our best regards,
The Authors
Reviewer 3 Report
Comments and Suggestions for Authors
The title succinctly outlines the primary aim of the study, emphasizing the dynamic relationship between biological activity and the dual factors of species variation and tissue specificity within sea cucumbers.
These results demonstrate a variable bioactive potential among sea cucumber species and tissues, suggesting a preference for targeting antioxidant properties in the muscle band and anti-inflammatory activity in the respiratory tree. How this information could serve as a foundational basis for a biorefinery approach?
Any storage and drying can affect the antioxidant potential of sea cucumber?
How Total polyphenol content was measured described complete methodology?
Overall, the findings presented in this paper offer a promising avenue for further exploration and potential utilization of sea cucumber bioresources in various fields.
Comments on the Quality of English LanguageCan be improved
Author Response
Dear Sir,
The whole paper was subjected to a new assessment in order to address the reviewer (and editor) concerns. For your advantage, we send you the revised manuscript with all alterations properly highlighted.
Reviewer #3:
The title succinctly outlines the primary aim of the study, emphasizing the dynamic relationship between biological activity and the dual factors of species variation and tissue specificity within sea cucumbers.
Thank you for your positive appraisal.
These results demonstrate a variable bioactive potential among sea cucumber species and tissues, suggesting a preference for targeting antioxidant properties in the muscle band and anti-inflammatory activity in the respiratory tree. How this information could serve as a foundational basis for a biorefinery approach?
This is an interesting line of analysis. It could be a step in the direction of a biorefinery approach. This approach would require further fractioning of each tissue, for instance, by isolating anti-inflammatory compounds in the respiratory tree of sea cucumbers. We understand that this reasoning would be helpful in the discussion of the results and, accordingly, this theme was addressed in the manuscript.
Any storage and drying can affect the antioxidant potential of sea cucumber?
This will be the target of a future study. This was added as suggestion into the Conclusions section of the paper.
How Total polyphenol content was measured described complete methodology?
The methodology for the determination of total polyphenol content was better described.
Overall, the findings presented in this paper offer a promising avenue for further exploration and potential utilization of sea cucumber bioresources in various fields.
Thank you for your positive appraisal.
Comments on the Quality of English Language
Can be improved
The whole manuscript’s text was revised by someone whose mother tongue is English.
We sincerely hope that the performed alterations are clear and acceptable. However, if needed, we will gladly take the changes in the manuscript even further.
With our best regards,
The Authors
Reviewer 4 Report
Comments and Suggestions for Authors
The manuscript New Biological Activity in Sea Cucumbers Changes as a Function of Species and Tissue aims to evaluate g phenolic content and the antioxidant and anti-inflammatory activity of aqueous extracts of four abundant species in the Portuguese coast (P. regalis, H. mammata, H. forskali, and H. arguinensis).
The introduction is clear and consistent, referring the most current publications in the field, pointing a specific gap of more profound knowledge of polyphenols content, antioxidant and anti-inflammatory potential of the specific tissues of Sea Cucumbers.
Materials and methods are described in details with appropriate references provided. Some clarifications are needed about the expression of for DPPH activity (L140-L143). The results are expressed as AAE (ascorbic acid equivalent), and compared with a Trolox positive control (100mg/l). However, in results section such a comparison is not provided. Furthermore in the referred method [23], the results are expressed in equivalent to the both standards used in the calibration activity (Trolox and vitamin C). In the presented work it is not clear what the Trolox positive control is applied to.
The results are well summarized with a proper statistical analysis of the data. The discussion is well structure and the results are compared with other literature sources and differentiation between tissues is examined in depth. А strong correlation between DPPH and FRAP results is observed in all sea cucumber samples.
The conclusion emphasizes the novelty of the findings, showing a variable bioactive potential in the sea cucumber species and their potential application.
Author Response
Dear Sir,
The whole paper was subjected to a new assessment in order to address the reviewer (and editor) concerns. For your advantage, we send you the revised manuscript with all alterations properly highlighted.
Reviewer #4:
The manuscript New Biological Activity in Sea Cucumbers Changes as a Function of Species and Tissue aims to evaluate g phenolic content and the antioxidant and anti-inflammatory activity of aqueous extracts of four abundant species in the Portuguese coast (P. regalis, H. mammata, H. forskali, and H. arguinensis).
Thank you for your appraisal.
The introduction is clear and consistent, referring the most current publications in the field, pointing a specific gap of more profound knowledge of polyphenols content, antioxidant and anti-inflammatory potential of the specific tissues of Sea Cucumbers.
Thank you for this positive assessment.
Materials and methods are described in details with appropriate references provided. Some clarifications are needed about the expression of for DPPH activity (L140-L143). The results are expressed as AAE (ascorbic acid equivalent), and compared with a Trolox positive control (100mg/l). However, in results section such a comparison is not provided. Furthermore in the referred method [23], the results are expressed in equivalent to the both standards used in the calibration activity (Trolox and vitamin C). In the presented work it is not clear what the Trolox positive control is applied to.
The missing information was included in the manuscript. More specifically, details about the calibration curve and the utilization of Trolox and the quantitative deviation between Trolox and samples considering its concentration and antioxidant activity were included.
The results are well summarized with a proper statistical analysis of the data. The discussion is well structure and the results are compared with other literature sources and differentiation between tissues is examined in depth. А strong correlation between DPPH and FRAP results is observed in all sea cucumber samples.
Thank you for your appraisal.
The conclusion emphasizes the novelty of the findings, showing a variable bioactive potential in the sea cucumber species and their potential application.
Meanwhile, the Conclusions section was improved in order to address some criticisms of the other reviewers. In particular, a future perspective section was included.
We sincerely hope that the performed alterations are clear and acceptable. However, if needed, we will gladly take the changes in the manuscript even further.
With our best regards,
The Authors